# Evidence for Range Expansion and Origins of an Invasive Hornet *Vespa bicolor* (Hymenoptera, Vespidae) in Taiwan, with Notes on Its Natural Status

**DOI:** 10.3390/insects12040320

**Published:** 2021-04-02

**Authors:** Sheng-Shan Lu, Junichi Takahashi, Wen-Chi Yeh, Ming-Lun Lu, Jing-Yi Huang, Yi-Jing Lin, I-Hsin Sung

**Affiliations:** 1Taiwan Forestry Research Institute, Council of Agriculture, Executive Yuan, Taipei City 100051, Taiwan; sslu@tfri.gov.tw (S.-S.L.); wcyeh@tfri.gov.tw (W.-C.Y.); 2Faculty of Life Sciences, Kyoto Sangyo University, Kyoto City 603-8555, Japan; jit@cc.kyoto-su.ac.jp; 3Endemic Species Research Institute, Council of Agriculture, Executive Yuan, Nantou County 552203, Taiwan; lily@tesri.gov.tw; 4Department of Plant Medicine, National Chiayi University, Chiayi City 600355, Taiwan; sars831212@gmail.com

**Keywords:** dispersal prediction model, hornet, invasive species, *Vespa bicolor*

## Abstract

**Simple Summary:**

The invasive hornet *Vespa bicolor* Fabricius was first discovered in Taiwan in 2003 and was not confirmed to have been established until 2014. This study was conducted in order to (1) assess the current status of *V. bicolor* abundance, dispersal, seasonality, and possible impact on honeybee (*Apis mellifera* Linnaeus) in Taiwan; (2) and to trace the origins of Taiwan’s *V. bicolor* population. To assess *V. bicolor* abundance, we used visual surveys, sweep netting, and hornet traps in four known ranges in northern and central Taiwan from 2016 to 2020. Additionally, to understand *V. bicolor* dispersion, we mapped environmental data using ArcGIS, and to predict future *V. bicolor* range, we used ecological niche modeling. The results show that *V. bicolor* has stable populations in three areas in northern and central Taiwan, and mainly preys on *Apis mellifera*. Our analyses suggest samples from Southeastern China as having the closest relation in DNA sequences with Taiwan’s *V. bicolor* population. Due to the negative economic and ecological impacts of *V. bicolor* in Taiwan, our findings shed light on the value of monitoring and controlling its populations, rather than working exclusively towards elimination.

**Abstract:**

The invasive alien species (IAS) *Vespa bicolor* is the first reported hornet that has established in Taiwan and is concerning as they prey on honeybee *Apis mellifera*, which leads to colony losses and public concerns. Thus, the aim of this study was to assess the current status of *V. bicolor* abundance, dispersal, and impact and to trace the origins of Taiwan’s *V. bicolor* population. Our studies took place in five areas in northern to central Taiwan. We used mtDNA in the phylogenetic analyses. Field survey and ecological niche modeling (ENM) were used to understand the origins and current range of the invasive species. Two main subgroups of *V. bicolor* in the phylogenetic tree were found, and a clade with short branch lengths in Southeastern China and Taiwan formed a subgroup, which shows that the Taiwan population may have invaded from a single event. Evidence shows that *V. bicolor* is not a severe pest to honeybees in the study area; however, using ENM, we predict the rapid dispersion of this species to the cooler and hilly mountain areas of Taiwan. The management of *V. bicolor* should also involve considering it a local pest to reduce loss by beekeepers and public fear in Taiwan. Our findings highlight how the government, beekeepers, and researchers alike should be aware of the implications of *V. bicolor*’s rapid range expansion in Taiwan, or in other countries.

## 1. Introduction

Invasive alien species (IAS) are non-native species that often cause adverse consequences in their new environment [1]. IAS should be monitored and controlled to prevent negative impact on agriculture and ecosystem function, as well as social and economic impacts on human beings [2]. The sheer diversity and niche breadth of insects allow them to play a significant role as IAS globally. Despite this, they are far understudied compared with plants and vertebrates. In addition, a recent review found that among the limited studies on IAS insects, nearly 18% were on the fire ant, *Solenopsis invicta* Buren, and up to 70% of the studies were conducted in North America [3]. These findings beg the need for studying IAS in Asia, especially in the North Asian region. Among insects, hornets are dominant and increasingly concerning players, as they can (1) be aggressively harmful to humans and other animals [4] and (2) spread easily due to nesting substrate availability during international trade [5,6].

Hornet has often become an example of an invasive species abroad. The yellow-legged hornet *Vespa velutina* Lepeletier is a widely distributed species indigenous to Southeast Asia, and it invaded Korea, Europe, and Japan [7,8,9,10,11]. Recently, there has been a lot of attention given to the giant Asian hornet *Vespa mandarinia* Smith, as it has invaded parts of North America [12]. This hornet species is the largest in Asia and has sparked public concern due to its painful sting and detriment to honeybee (*Apis mellifera* Linnaeus) populations. Hornet belongs to the family of social wasps in the genus *Vespa* (family: Vespidae), which comprises 22 species, all which were restricted to Asia until introduction and invasions occurred [13]. These species are important for the control of other insects in their native range [14], yet can be destructive to those outside it; thus it is critical to understand the biology and status of local species to avoid such perils.

In 2003, Taiwan first reported the invasive hornet species *Vespa bicolor* Fabricius [15], known as two-colored hornet or black shield hornet in Chinese [16,17]. *V. bicolor* is widely distributed in India, Bhutan, Cambodia, China, Laos, Myanmar, Nepal, Thailand, and Vietnam [18]. The existence of *V. bicolor* in Taiwan is concerning as it is known to prey on *A. mellifera*, which leads to colony loss and economic unfavorability [15]. Numerous hornets of the *Vespa* spp. have been reported to prey on honeybees in Asia [19,20], such as the hornets *Vespa affinis* (Linnaeus), *Vespa basalis* Smith, *V. bicolor*, *Vespa ducalis* Smith, *V. mandarinia*, and *Vespa velutina* in Taiwan [15]. These hornets are very adaptable and exhibit a wide range of prey capture strategies, from group hunting to sit and wait [4,15]. In addition, hornets, honeybees, and honey buzzards evolve complex interactions between prey and predator. For example, the migrant Oriental honey-buzzard *Pernis ptilorhynchus orientalis* Taczanowski preys on hornets (*V. velutina*), wasps (*Polistes* spp., Vespidae), and honeybees (*A. mellifera*) [21]; and a few populations are adapting to become summer residents in Taiwan [21,22]. Besides, the European honey buzzard *Pernis apivorus* (Linnaeus) has adapted to prey on hornet *V. velutina* [23].

Knowledge of IAS origin through the analysis of DNA and ecological landscape structures can help decision makers to draft risk assessment and management strategies [24,25]. The studies used mitochondrial and microsatellite DNA for *V. velutina* samples, and the origins of Japan, South Korea, and Europe populations were traced [10,11,26]. Furthermore, an intraspecies genetic study on *V. velutina nigrithorax* du Buysson that tested microsatellite variation in colonizing populations that were introduced via a very small number of propagules provided valuable information [26]. Moreover, ecological niche modeling (ENM) has been widely used for predicting the spread of IAS and for simulating spatial distribution. Using ENM and data from native and invaded ranges to assess the potential invasion risk of the invasive hornet *V. velutina nigrithorax* has predicted a noticeable remark on beekeeping activities that are threatened factors [27].

This study followed our 2014 report of *V. bicolor* establishment in Taiwan [15]. We used phylogenetic analyses, field survey, and ENM to understand the origins and current range of the invasive species. For the phylogenetic analyses, we sequenced mitochondrial DNA from *V. bicolor* samples to compare the genetic variations across their range. For the field survey, we conducted proactive monitoring of the *V. bicolor* activities to know the current temporal and spatial occurrences. For the ENM, the geographical locations were included in the model. These records contained long-term field surveys: (1) collection records from 2011 to 2013 of our study [15], (2) intermittent notifications of *V. bicolor* activity ranges from beekeepers or civilians from 2014 to 2020, (3) and survey data in this study from 2016 to 2020. We intend for our study to inform future control and management strategies.

## 2. Materials and Methods

### 2.1. Genetic Variations across V. bicolor Ranges

Since previous reports of *V. bicolor* hornets were predominantly from apiaries or from their own nests in northern and central Taiwan, the samples were collected for phylogenetic analyses there. The collections were carried out in Taoyuan City and the counties of Hsinchu and Miaoli from 2016 to 2020 discontinuously (Table 1). We collected a total 23 female individuals from different locations, 7 from the apiaries, 2 from flowers, and 14 from their own nests (Table 1). The samples were preserved in 95% EtOH and stored at 4 °C for subsequent DNA analysis.

The genomic DNA was extracted from a single midleg of the *V. bicolor* samples using an DNA Extraction Kit (Fisher Biotec, Wembley, Western Australia). The target DNA was amplified using the universal LCO1490/HCO2198 primers to get the COI region [28]. However, not all samples were successful in using the recommended protocol with LCO/HCO primers, as mentioned in the studies [29,30]. We also used the primers suggested by van Houdt, L1440d (5- TYT CAA CWA ATC ATA ARG ATA TTG G -3) and H2123d (5- TAW ACT TCW GGR TGW CCA AAR AAT CA -3), which can improve the amplification efficiency [30]. The primers used in the amplification were effective for our difficult samples, and we obtained the same fragment of the COI region [31].

Polymerase chain reaction (PCR) was performed with a thermocycler (Applied Biosystems 2720 Thermal Cycler, Foster, CA, USA). Each PCR reaction mixture contained 0.5 µL of Vas Taq (5 units/µL) (Bionovas Biotechnology Co., Ltd., Toronto, Canada), 5 µL of 10× Taq buffer (Bionovas Biotechnology Co., Ltd., Toronto, Canada), 4 µL of 2.5 mM dNTPs (Won-Won Biotechnology Co., Ltd., New Taipei City, Taiwan), 1 µL of 10 µM forward and reverse primers, 1 µL of template DNA, and deionized water (up to 50 µL). The PCR amplification consisted of an initial denaturation at 94 °C for 5 min, followed by 40 cycles of denaturation at 94 °C for 30 s, annealing at 50 °C for 30 s, extension at 72 °C for 30 s, and final extension at 72 °C for 10 min. PCR products were separated by electrophoresis in 1% agarose gel, stained with ethidium bromide, and visualized with a UV transilluminator (Major Science, CA, USA) The genes were sequenced with Sanger dideoxy sequencing by a commercial genome sequencing company (Genomics, New Taipei City, Taiwan).

The software MEGA X was used to compare [32], align, and trim the double-stranded base pair sequences of the same arrangement of COI sequences. The supplemental sequences from Huangshan in China, Hong Kong, Nepal, and Vietnam were provided by one of our authors (J.T.). We also included sequence data on *V. bicolor* from GenBank (accession nos. KT257112, KF933079, Vietnam; KJ735511, Beijing) [33] and used as outgroups [34] *Vespa simillima* (accession no. NC046020), *V. velutina* (accession no. NC035146), and *V. mandarinia* (accession no. KR059904). The model of maximum likelihood method and Tamura 3-parameter, 1000 bootstrap numbers were used to construct phylogenetic trees. DnaSP v6 software was used to detect *V. bicolor* COI haplotypes [35]. For this analysis, we also included data on COI sequences from GenBank (*V. bicolor* accession nos. KT257112, KF933079, Vietnam; KJ735511, Beijing). The haplotype network was calculated using Network v10 and plotted by hand [36].

### 2.2. Occurrences, Daily and Seasonal Activities of V. bicolor

The studies took place in five areas from northern to central Taiwan: the counties Hsinchu, Miaoli, and Nantou and the cities Taoyuan and Taichung (Figure 1). Investigation by visual surveys was carried out at 13 apiaries in the cities of Taoyuan and Taichung and the counties of Hsinchu, Miaoli, and Nantou from April to November discontinuously in 2019 and 2020 (Table 2). In addition, 7 apiaries were set up with hornet traps, 1 to 2600 cc PET bottles with two entrances and filled 50% with alcohol-based bait attractant (Table 2) [37]. Moreover, beekeepers and affiliated persons were trained to distinguish *V. bicolor* hunting behaviors and activities, and we recorded their findings and notifications. The apiary (ca. 30 hives) of *V. bicolor* occurred in Tongluo, Miaoli, where we used a visual survey monthly from May to November in 2019 and 2020. Each *V. bicolor* count lasted for 1 h by two researchers. In addition, the apiaries (each apiary with 30 hives) of *V. bicolor* were observed to occur in Tongluo, Miaoli, and Beipu, Hsinchu, using mark and recapture for 5 days (twice a day) during the peak occurrence of the year (i.e., from September to October 2020). Each survey was done by two researchers for 1 h, and the interval was 2 h between the first and second survey. The *V. bicolor* captured during the first survey was marked with a color acrylic pen marker, and then the *V. bicolor* captured during the second survey, and the number of captured was counted and recorded. The daily and seasonal activities of *V. bicolor* at the apiary were estimated.

### 2.3. Ecological Niche Modeling

#### 2.3.1. Species Occurrence and Environmental Data

Twenty-seven geographical locations in the counties and cities of Taoyuan, Hsinchu, Miaoli, and Taichung, where *V. bicolor* was most likely to occur, were included in the model (Figure 1). Environmental predictors refer to the critical factors that affect species distribution. To determine which environmental variables influence the distribution of *V. bicolor*, we included in our model 19 bioclimatic variables [38] and 1 biophysical variable (elevation) with a 30 s (ca. 1 km) spatial resolution, downloaded from a WorldClim dataset (www.worldclim.org) (accessed on 3 November 2020). The mean slope and aspect were calculated from the elevation layer using a surface analysis tool from the Spatial Analyst toolbox in ArcGIS 10.6 (Esri) (Environmental Systems Research Institute, Redlands, CA, USA). Then, we used the Geomorphometry and Gradient Metrics toolbox (version 2.0) [39] to generate compound topographic indices (representing humidity) and heat load indices (representing solar radiation intensity). To reduce the multicollinearity among the 24 variables (Appendix A), the variance inflation factor (VIF) implemented in the “usdm” package in the R platform version 3.6.3 was used to exclude predictors with VIF values >10 [40].

#### 2.3.2. Ensemble Modelling

We employed an ensemble modelling approach to predict *V. bicolor* distribution using R with the “sdm” package [40]. We selected five algorithms: generalized linear models (GLMs), multiple adaptive regression splines (MARS), random forest (RF), maximum entropy (MaxEnt), and support vector machine (SVM). As all these models require background data (pseudo-absence points), we generated a random sampling of 5000 locations across the study area. Models were evaluated using *k*-fold cross-validation with 10 folds and 10 replications for each algorithm. We projected each of the models using 70% of the training data and 30% for evaluation. The performance of the models was evaluated using threshold-independent indices of the area under the curve (AUC) of a receiver operating characteristic and the true skill statistic (TSS) [41,42]. The AUC ranged from 0.5 to 1, with a higher value denoting greater prediction accuracy. An AUC value ≥0.7 represents reliable prediction accuracy [43]. The relative importance of predictor variables was estimated using the function getVarImp of the “sdm” package. Moreover, to obtain the consensus predictions, we used the function “ensemble” based on weighted TSS values. Finally, the output map represents the occurrence probability for each pixel. We extracted the pixels with maximum occurrence probability, representing probabilities of more than 75% as suitable habitats [44].

## 3. Results

### 3.1. Genetic Variations across V. bicolor Ranges

The phylogenetic tree of the COI region near the 5′ end showed that all *V. bicolor* hornets from Beijing and Huangshan, Hong Kong, Nepal, Taiwan, and Vietnam were closely related to form a monophyletic clade (Figure 2A). Two main subgroups diverged: a clade with short branch lengths in Huangshan, Hong Kong, and Taiwan formed the first subgroup, while the Nepal samples and a clade in Beijing and Vietnam formed the second. All individuals collected in Miaoli, Hsinchu, and Taoyuan in Taiwan have identical sequences. The Beijing and Vietnam group had a relatively distant relationship with the Nepal group. The phylogenetic results show a *V. bicolor* population in Taiwan, suggesting it may come from the same source near Huangshan and Hong Kong in the Southeastern China area. The Taiwan sequences were submitted to GenBank (accession nos. MW455061–MW455084) (Table 1). Moreover, we found seven haplotypes in the COI sampling of *V. bicolor* (from Hap_1 to Hap_7) (Figure 2B and Appendix A). All Taiwan samples were detected in haplotype Hap_3, with three mutations linked to Hong Kong samples (Hap_5) and Huangshan samples (Hap_6) to form a branch. The second and third branches were Beijing samples (Hap_2) and Nepal samples (Hap_4), respectively. The Vietnam haplotype includes Hap_1 and Hap_7, which was linked to two median vectors. The haplogroups radiate from Hap_1 and Hap_7 to form a tripodlike pattern, indicating that the Vietnam samples represent ancestral haplotypes.

### 3.2. Current Status, Daily and Seasonal Activities of V. bicolor

The field survey recorded six hornet species, *V. affinis*, *V. basalis*, *V. bicolor*, *V. ducalis*, *V. mandarinia*, and *V. velutina*, in 13 apiaries in the city of Taoyuan and the counties of Hsinchu, Miaoli, and Nantou. The body size of *V. bicolor* (15–19 mm) is smallest, but it is a fast-flying hornet among these *Vespa* species, so it is not easy to spot and catch around the hives. According the beekeepers and affiliated persons’ notifications, a native Taiwanese wasp, *Polistes rothneyi* Cameron, was easy to confuse with *V. bicolor* in body color and size, which gave false information. There were 125 notifications of *V. bicolor* from 2016 to 2020, of which 82 (65.6%) notifications were invalid records. A total of 43 (34.4%) notifications were recorded as *V. bicolor* (Figure 3A), and most of them were individuals. Among them, 12 nests were located and removed (Figure 3B). The number and proportion of these notifications by city and county included 2 in Taoyuan (4.6%), 11 in Hsinchu (25.6%), and 30 in Miaoli (69.8%). We also confirmed that 6 of 13 apiaries in the city of Taoyuan and the counties of Hsinchu and Miaoli had *V. bicolor* records (Table 2). Seven apiaries failed to confirm *V. bicolor* occurrences in our monitoring, where there were 4 apiaries in the city of Taichung and the county of Nantou. Based on beekeeper witnesses, there was 3 apiaries in the counties of Hsinchu and Miaoli and the city of Taichung (Table 2). The years of these findings by city and county included 2020 (Taoyuan), 2016–2020 (Hsinchu), 2011–2020 (Miaoli), and 2019 (Taichung). Until 2020, the occurrences of *V. bicolor* confirmed by our research team were in the city of Taoyuan and the counties of Hsinchu and Miaoli.

There were 6 *V. bicolor* nests removed in Hsinchu County, 5 in Miaoli County, and 1 in Taoyuan City. There were 3 nests removed in Hsinchu and Miaoli from 2016 to 2019. However, in 2020, the number of removals reached 9 nests in Taoyuan, Hsinchu, and Miaoli (Figure 3A), showing a rapidly increasing trend. The months for the removal were mainly from July to November, but 2 nests were discovered in January 2019 (Figure 3B). Among the nest sites, 7 of 12 (58%) nests were built in the tree holes of *Acacia confusa* Merr., *Cinnamomum camphora* (L.) J. Presl, *Cunninghamia* sp., and *Lagerstroemia subcostata* Koehne. There were 3 of 12 (25%) nests that were built in man-made places, such as discarded tires, abandoned water towers, and wooden loudspeakers. There were 2 of 12 (17%) nests that were built in underground caves formed by the root spaces of plants (Figure 3C). Most of the nests in tree holes and underground caves cannot be taken out completely, and can only be removed with insecticides or by blocking the entrance of the nest. The adults in 9 nests were caught and counted as much as possible, which contained 29 to 1030 individuals.

Based on visual surveys, seasonal activities of *V. bicolor* were found from July to October at the Tongluo apiary in 2019 and 2020 (Figure 3D). The peak activities in the Tongluo and Beipu apiaries were from September to October in 2020, and there were nine active surveys during sunny, cloudy, and rainy days, with the exception of a single day due to rain conditions at Beipu on 22 October 2020 (Figure 3E). The number of *V. bicolor* hornets that attacked honeybees ranged from 1 to 28 per hour; in addition, it was found that the marked *V. bicolor* was captured in the second investigation at the Tongluo apiary (Figure 3E). The mean hourly numbers and standard deviations of *V. bicolor* hunters were about 9.8 ± 8.1 in Tongluo and 4.4 ± 2.9 in Beipu.

### 3.3. Potential Range for V. bicolor

The 24 initial variables were reduced to 8 after applying the standard of VIF values >10. The obtained values of TSS and AUC for five modelling algorithms indicated good predictive performance (TSS, >0.71 and AUC, >0.83), which can be considered an accurate validation. The results showed that the selected variables described the distribution of *V. bicolor* well. Among the eight environmental variables (Appendix A), temperature seasonality (BIO4, 48.8% contribution) and temperature of driest quarter (BIO9, 12.8% contribution) were the two most important predictors determining the distribution (Appendix A).

The total amount of pixels with maximum occurrence probability (suitable habitats) covered 440 km^2^. This area was mainly distributed to cover 232 (52.0%) and 168 (38.0%) km^2^ in the counties of Miaoli and Hsinchu (Figure 4). In addition, these areas were distributed in hilly or shallow mountains below 1000 m above sea level (Figure 4).

## 4. Discussion

Biological invasions often cause improper affection to the natural ecological environment, such as the decline of biodiversity or deterioration of ecosystem functions [45,46,47], threat to the native flora [48], and threat to agricultural and urban environments [49,50,51], and the levels involved can also harm the human and social systems [52]. Additionally, they posed a risk to aviation safety in Australia recently [53]. Due to multiple factors, hornet has become an invasive alien species (IAS), not only causing harm to the ecological environment, but also bringing a lot of challenges to human society, economy, and agricultural production safety. Since hornets have no nesting preference, they often establish colonies in anthropogenic areas, posing a potential threat to humans and other animals. Every year in Taiwan, the number of treated patients who experience Vespidae stings has been greater than those who experience Formicidae stings, causing life hazards [54]. To prevent this spread of the wasp species, government officials often adopt by incentivizing elimination programs, or hiring hornet hunters to collect nests for hornet liquor or larva dishes [55]. However, hornets are omnivores that can use various animal body or botanical products as their food, such as honeybees, rotten fruits, and plant nectars [14]. They hunt both beneficial insect pollinators and pest species, such as those in the orders Diptera and Hemiptera [14]. They have also been found to be a biological control for *Polistes* wasps [56]. We considered these elimination actions to be contentious due to the potential for negative ecological impact or harm to those proximal to the removal effort.

Although there were few samples out of Taiwan in this study, according to the phylogenetic analysis, the *V. bicolor* collected from different regions of Taiwan have high similarity. Only the samples from Southeastern China had high similarity with those from Taiwan. The database of this species in GenBank was not complete, so we can infer from the current results that the invasion of *V. bicolor* in Taiwan may have resulted from a single event. The earliest record of this species in Taiwan can be traced back to 2003 in Taichung City according to personal observations by J. T. Chao [15]. In previous studies concerning the Taiwanese hornet fauna, *V. bicolor* was never mentioned [57,58]; thus we reasonably confirmed our previous findings [15].

It was found that, until 2020, the population of *V. bicolor* was quite stable in the counties of Hsinchu and Miaoli. Miaoli is a county where *V. bicolor* has steadily occurred in all discovered regions, and monitoring has been done since 2011. Since 2016, Hsinchu County has noticed *V. bicolor* activities in apiaries, and nests have been occasionally found. In 2019 and 2020, there was evidence of nest activities and dispersal to Taoyuan City, and a suspected case was found in Taichung. The result shows that the trend of occurrence of *V. bicolor* is developing towards the northern or central areas of Taiwan. In Hong Kong, *V. bicolor* is widely distributed, and the activity period and colony cycle are extremely long and very tolerant of cold weather [17]. Moreover, nests were built in all kinds of environments, including in underground crevices and in and around buildings, all sorts of assorted objects, small trees, and shrubs [17]. These situations are similar to our findings.

During the peak activity in a suitable weather, the number of *V. bicolor* preying on honeybees was up to 28 at the Tongluo apiary per hour. The average number ranged from 4.4 to 9.8 per hour at the Tongluo and Beipu apiaries. An individual *V. bicolor* hornet was reported to prey on honeybees twice a day in the peak activity season. Previous studies found that the activity range of *V. velutina* depended on resources, but could expand up to 1 km from the nest [59]. Here we find that food resources and altitude are the two most important environmental factors determining the location of *V. bicolor* nests. The predation behavior of *V. bicolor* is similar to that of *V. velutina* [15]. The number of *V. bicolor* individuals was between 45 and 98 per day (for a 10 h activity calculation). Apiaries were reported to have up to 350 *V. velutina* individuals per day [60], indicating that *V. bicolor* has not been an important pest to honeybees until now. Therefore, the relevant data could provide some reference. This study provides a preliminary understanding of the situation of *V. bicolor* preying on honeybees.

With respect to ENM, the obtained AUC and TSS values were all larger than 0.7, which indicates that the predicted results were reliable [45,61]. We assessed the contributions of various environmental variables in the prediction of the *V. bicolor* distribution, and results indicate that temperature seasonality and climate were the key factors of species spatial distributions. Previous studies also found that the effects of temperature on hornets were significantly greater than those of other environmental factors [62]. The ecological niche of *V. bicolor* can greatly aid our understanding of changes to their geographical ranges. According to the result of the simulation, *V. bicolor* is distributed in the hilly or shallow mountain areas in Hsinchu and Miaoli counties. These areas, potentially due to having a suitable climate, provide nesting spaces and food resources for *V. bicolor*.

Our study found that there is not much genetic variation in the known existing populations. In fact, in the known positions of the existing records, the shortest distance between two points is not far away (not more than 15 km). Based on the estimation of the longest foraging distance of *V. velutina*, it is 11 km [63]. We suppose that there is no significant genetic variation in the existing *V. bicolor* populations. The reason is either there is a single source of invasion or there is no geographical isolation or both. In the future, more information can be learned from a large population of nuclear microsatellite DNA analysis [26]. Relevant information will be carried over to future experiments.

## 5. Conclusions

In the prediction of the dispersal model and monitoring survey, it was found that Hsinchu and Miaoli had stable *V. bicolor* populations due to the climate condition in these counties. According to the prediction results, the suitable habitat for *V. bicolor* is about 440 km^2^. Additionally, many local populations spread to the northern area of Taoyuan City; thus we assess that *V. bicolor* adapts to cooler weather in subtropical areas. In addition, the geographical distribution range was from hilly to shallow mountain areas of Taoyuan, Hsinchu, and Miaoli, potentially due to increased spaces and resources for nesting. Our results show that the *V. bicolor* population in Taiwan might have derived from the same invasion source. The predation of *V. bicolor* was mainly on honeybees, and the range of forage activities was highly connected to the presence of apiaries. ENM techniques allowed the spread of the invasive species after its recent arrival to be predicted with a good level of accuracy; hence, we recommend focusing on control efforts in these areas. Although the continuous removal of the IAS is often necessary, the management of *V. bicolor* should also involve considering it a local pest to reduce loss by beekeepers and public fear. Our findings highlight that we should be aware of the *V. bicolor’s* rapid range expansion in Taiwan. The value of monitoring and controlling its populations brings to light further control plan improvement.

## Figures and Tables

**Figure 1 insects-12-00320-f001:**
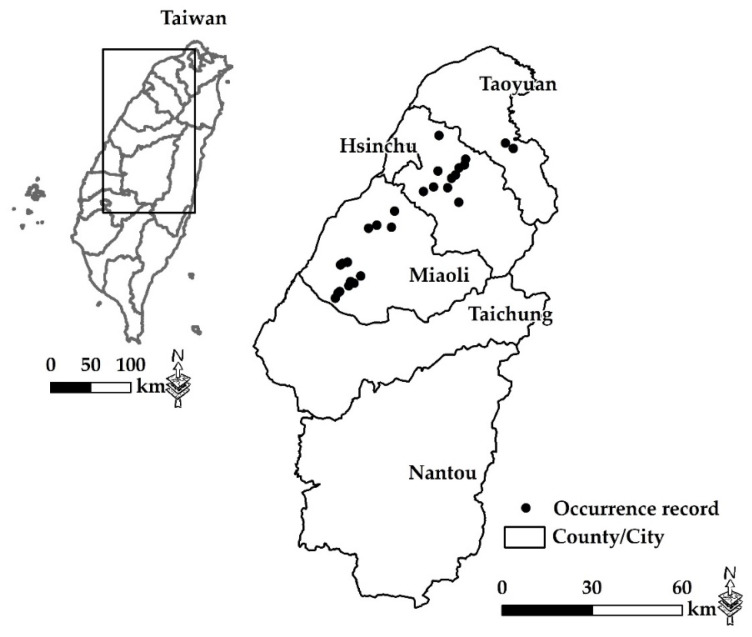
Occurrence records of *V. bicolor* in Taiwan.

**Figure 2 insects-12-00320-f002:**
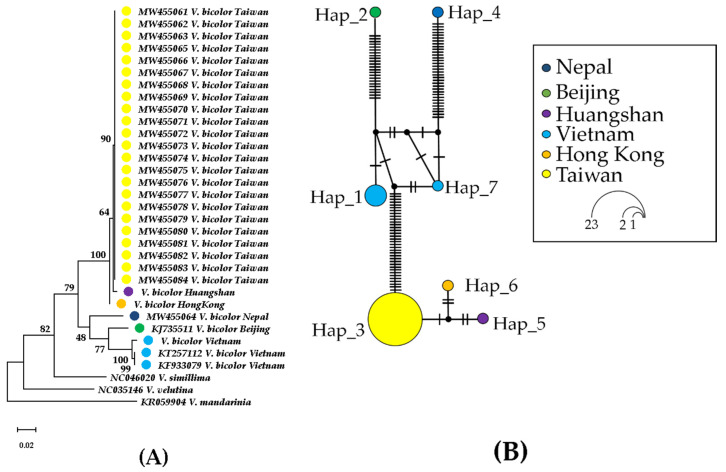
Phylogenetic tree and haplotype network of *V. bicolor* from China, Hong Kong, Nepal, Taiwan, and Vietnam. (**A**) Phylogenetic tree computed on COI sequences. Each number on the branch represents node stats in maximum likelihood bootstrap values. (**B**) Haplotype network. Each bar on the branch represents a single nucleotide substitution. Circle size is equivalent to sample size. Median vectors are represented by black circles.

**Figure 3 insects-12-00320-f003:**
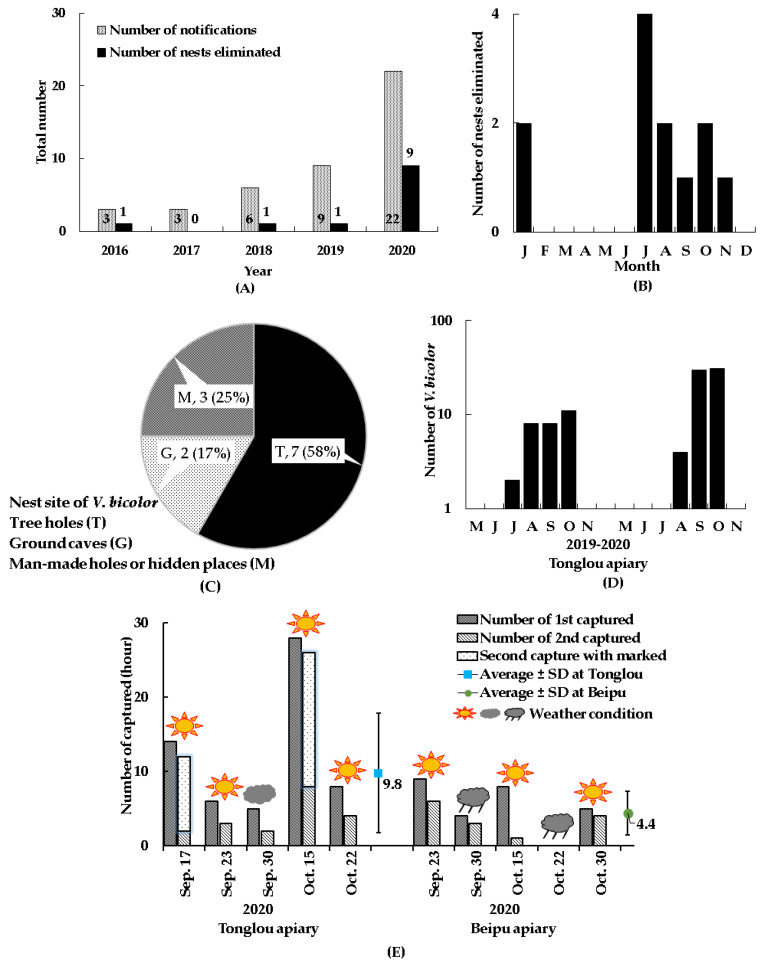
Field survey of *V. bicolor* from Taiwan: (**A**) numbers of notifications and eliminated nests of *V. bicolor*; (**B**) months during which nests were eliminated; (**C**) nest site, number of nests, and proportion in *V. bicolor*; (**D**) months during which *V. bicolor* activities were observed; (**E**) daily activities of *V. bicolor* in the Tongluo and Beipu apiaries.

**Figure 4 insects-12-00320-f004:**
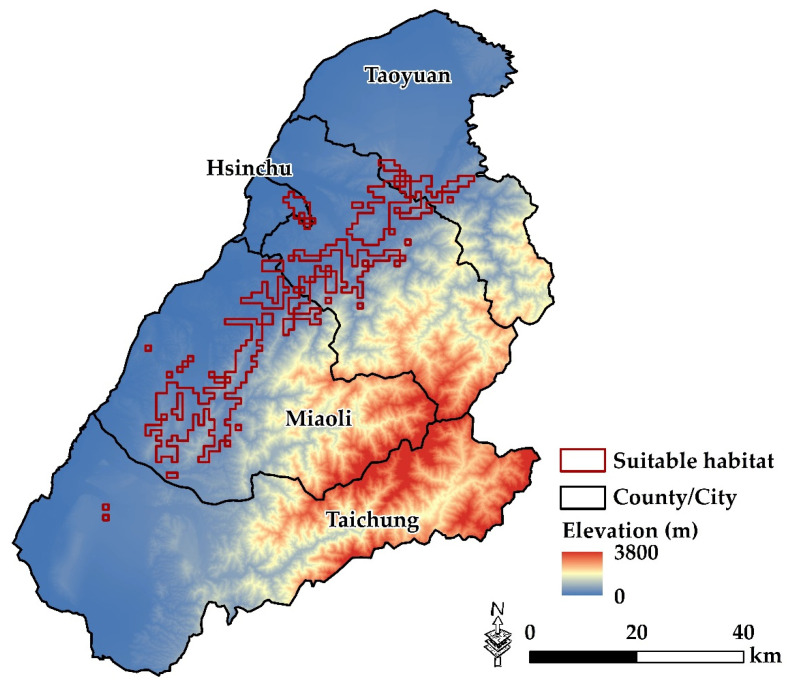
Prediction of potential distribution area for *V. bicolor* in Taoyuan, Hsinchu, Miaoli, and Taichung areas.

**Table 1 insects-12-00320-t001:** The sample of *Vespa bicolor* from Taiwan, with the collection site and accession number in GenBank.

Species City/County	Locality	Year, Date	Collection Site			GenBank Accession Number
			Attacking Bees in Apiaries	Visiting Flowers	Own Nests	
*Vespa bicolor*						
Taoyuan	Daxi	17 July 2020	○			MW455072
	Fuxing	29 August 2020			○	MW455074
Hsinchu	Beipu	15 November 2019	○			MW455066
	Beipu	17 July 2020			○	MW455071
	Guanxi	22 July 2019			○	MW455065
	Guanxi	10 August 2020			○	MW455073
	Hengshan	22 October 2020	○			MW455078
	Hengshan	13 January 2016			○	MW455082
	Hengshan	13 January 2016			○	MW455083
	Qionglin	4 November 2018			○	MW455081
	Wufeng	9 September 2020			○	MW455076
	Xinpu	30 October 2020	○			MW455079
	Zhudong	3 January 2020			○	MW455068
Miaoli	Sanwan	16 November 2017		○		MW455084
	Sanyi	3 January 2020			○	MW455067
	Tongluo	16 August 2019	○			MW455061
	Tongluo	2 August 2019		○		MW455062
	Tongluo	18 September 2019	○			MW455063
	Tongluo	16 July 2020			○	MW455069
	Tongluo	16 July 2020			○	MW455070
	Tongluo	19 August 2020	○			MW455075
	Tongluo	30 October 2020			○	MW455080
	Touwu	30 October 2020			○	MW455077

○ confirmed in this study.

**Table 2 insects-12-00320-t002:** The field survey and occurrence records of *V. bicolor* in 13 apiaries.

City/County	Locality	Survey Year	Hornet Trap	Occurrence
Taoyuan City	Daxi	2020	No	○
	Fuxing	2020	No	○
Hsinchu County	Beipu	2019–2020	Yes	○
	Guanxi	2019	Yes	△
	Hengshan	2020	No	○
	Xinpu	2020	Yes	○
Miaoli County	Dahu	2020	No	△
	Tongluo	2019–2020	Yes	○
	Tongxiao	2019	Yes	—
Taichung City	Houli	2020	No	—
	South District	2020	Yes	△
	Wufeng	2020	Yes	—
Nantou County	Guoxing	2020	No	—

○ confirmed in this study; — not confirmed in this study; △ beekeeper records.

## Data Availability

All data presented in this study are available in the article.

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
