# Peer review of "Evidence for Range Expansion and Origins of an Invasive Hornet Vespa bicolor (Hymenoptera, Vespidae) in Taiwan, with Notes on Its Natural Status"

_insects, 2021, doi:10.3390/insects12040320_

Round 1
Reviewer 1 Report
- Abstract. You need to increase the abstract.
- Introduction. In the first paragraph, it is necessary to add several publications on the topic of invasive and actively distributed species of Hymenoptera (https://journals.plos.org/plosone/article?id=10.1371/journal.pone.0185172; Proshchalykin M.Yu., Sergeev M.E. 2020. New distribution data of Apis cerana ussuriensis (Hymenoptera, Apidae) from Primorsky Krai, Russia. Nature Conservation Research 5(4): 111–112. https://dx.doi.org/10.24189/ncr.2020.049; Ruchin A.B., Artaev O.N. On expansion of the distribution range of some scoliid wasps (Scoliidae, Hymenoptera, Insecta) in the Middle Volga region. Research Journal of Pharmaceutical, Biological and Chemical Sciences. 2016. V. 7. Is. 3. P. 2110-2115.; https://onlinelibrary.wiley.com/doi/abs/10.1111/icad.12418). Authors should write about invasive species that have invaded Europe and Asia.
- Materials and Methods. They are presented adequately.
- Results. The results are presented clearly.
- Discussion. Line 293-294. Text "Biological invasions often cause improper affection to the natural ecological environment, such as the decline of biodiversity or the deterioration of ecosystem functions [38], and the levels involved can also harm the human and social system [39]." This sentence should be enlarged and clarified by the example of plants, animals, and other Insecta. Here are the publications to cite (https://dx.doi.org/10.24189/ncr.2019.013; DOI: 10.1007/978-94-007-7750-7; DOI: 10.1111/gcb.13798; Ruchin A.B., Egorov L.V., Lobachev E.A., Lukiyanov S.V., Sazhnev A.S., Semishin G.B. 2020. Expansion of Harmonia axyridis (Pallas, 1773) (Coleoptera: Coccinellidae) to European part of Russia in 2018–2020. Baltic J. Coleopterol., 20 (1): 51–60.; http://dx.doi.org/10.24189/ncr.2017.044). b) Line 295-296: It is necessary to clarify who gave such information.
- Conclusions. It is necessary to clarify what the authors propose to do to reduce the hornet infestation in Taiwan.
Author Response
We thank the reviewer for their appraisal of the work, and are pleased that the description of the result is well understood.
Thank you for those suggestions and relevant literature provided, we will add in our article content properly as list below.
Abstract
[...] "Thus, this study was to assess the current status of V. bicolor abundance, dispersal, impact and to trace the origins of Taiwan's V. bicolor population. [...]
Introduction [...] Examples from insect, either the new distribution Apis cerana or range expansion of some scoliid wasps need further study. [3, 4]. [...]
Hornets has often become an example of an invasive species abroad. Yellow-legged horent Vespa velutina Lepeletier is a widely distributed species indigenous to Southeast Asia, and it invaded to Korea, Europe and Japan [9–13].
Discussion [...] Biological invasions often cause improper affection to the natural ecological environment, such as the decline of biodiversity or the deterioration of ecosystem functions [47–49], threat for the native flora [50], and the levels involved can also harm the human and social system [51]. And even more, the risk to aviation safety in Australia recently [52][...]
Conclusions [...] Our findings highlight we should be aware of the V. bicolor’s rapid range expansion in Taiwan. The value of monitoring and controlling their populations bring light to further control plan improvement [...]

Reviewer 2 Report
These are my main comments on the MS (insects-1147798) entitled: "Evidence for range expansion and origins of an invasive hornet Vespa bicolor (Hymenoptera, Vespidae) in Taiwan, with notes on its natural status" by Lu et al.
My main objections are about the ENMs:
L175-185: What is the extent of your study? Did you perform any data reduction analysis to reduce the number of climatic variables you used here? Using all 19/23 for each scenario is not good at all! Beaumont et al. 2005 is one of the references I recall that evaluated this matter, but there are several others as I show you below. Later I saw you removed variables considering “highly correlated variables” (L185). What were the “highly correlated variables”? please explain.
Beaumont, L. J., Hughes, L., & Poulsen, M. (2005). Predicting species distributions: Use of climatic parameters in BIOCLIM and its impact on predictions of species’ current and future distributions. Ecological Modelling, 186(2), 251–270. https://doi.org/10.1016/j.ecolmodel.2005.01.030
Dormann, C. F., McPherson, J. M., Araújo, M. B., Bivand, R., Bolliger, J., Carl, G., Davies, R. G., Hirzel, A., Jetz, W., Kissling, D., Köhn, I., Ohlemüller, R., Peres-Neto, P. R., Reineking, B., Schröder, B., Schurr, F. M., & Wilson, R. (2007). Methods and to account for spatial autocorrelation in the analysis of species distributional data: a review. Ecography, 30, 609–628.
Dormann, C. F., Schymanski, S. J., Cabral, J., Chuine, I., Graham, C., Hartig, F., Kearney, M., Morin, X., Römermann, C., Schröder, B., & Singer, A. (2012). Correlation and process in species distribution models: Bridging a dichotomy. Journal of Biogeography, 39, 2119–2131. https://doi.org/10.1111/j.1365-2699.2011.02659.x
De Marco, P., & Nóbrega, C. C. (2018). Evaluating collinearity effects on species distribution models: An approach based on virtual species simulation. PLoS One, 13(9), e0202403. https://doi.org/10.1371/journal.pone.0202403
L188-198: There are better ways of building your training/testing subsets than random 70by30% partition.
Roberts, D. R., Bahn, V., Ciuti, S., Boyce, M. S., Elith, J., Guillera-Arroita, G., Hauenstein, S., Lahoz-Monfort, J. J., Schröder, B., Thuiller, W., Warton, D. I., Wintle, B. A., Hartig, F., & Dormann, C. F. (2017). Cross-validation strategies for data with temporal, spatial, hierarchical, or phylogenetic structure. Ecography, 40(8), 913–929. https://doi.org/10.1111/ecog.02881
L1194-198: AUC is a very poor metric to evaluate the model’s results, and it is highly criticized (Lobo et al. 2008; Jiménez-Valverde 2012, 2014). There are better metrics available in the market (Leroy et al. 2018).
Leroy, B., Delsol, R., Hugueny, B., Meynard, C. N., Barhoumi, C., Barbet-Massin, M., & Bellard, C. (2018). Without quality presence–absence data, discrimination metrics such as TSS can be misleading measures of model performance. Journal of Biogeography, 45(9), 1994–2002. https://doi.org/10.1111/jbi.13402
Lobo, J. M., Jiménez-Valverde, A., & Real, R. (2008). AUC: a misleading measure of the performance of predictive distribution models. Global Ecology and Biogeography, 17(2), 145–151. https://doi.org/10.1111/j.1466-8238.2007.00358.x
Jiménez-Valverde, A. (2012). Insights into the area under the receiver operating characteristic curve (AUC) as a discrimination measure in species distribution modelling. Global Ecology and Biogeography, 21(4), 498–507. https://doi.org/10.1111/j.1466-8238.2011.00683.x
Jiménez-Valverde, A. (2014). Threshold-dependence as a desirable attribute for discrimination assessment: Implications for the evaluation of species distribution models. Biodiversity and Conservation, 23(2), 369–385. https://doi.org/10.1007/s10531-013-0606-1
L187-198: What was the threshold used to generate presence/absence maps? What were the future climatic scenarios used to generate future distributions and their sources? These are only two problems I could raise in a diagonal reading of the text. I am sure others may be found. Also, did you know that Maxent is very criticized if used alone, and several other improvements need to be taken care of Maxent’s settings and parametrization before you run it? Please see Muscarella et al. 2014 for some of these issues. I stress out that the authors need to acquire a good SDM text and check its M&M section to better structure the M&M section’s text.
Muscarella, R., Galante, P. J., Soley-Guardia, M., Boria, R. A., Kass, J. M., Uriarte, M., & Anderson, R. P. (2014). ENMeval: An R package for conducting spatially independent evaluations and estimating optimal model complexity for Maxent ecological niche models. Methods in Ecology and Evolution, 5(11), 1198–1205. https://doi.org/10.1111/2041-210X.12261
The next draft of this paper will need to be dramatically different to have a chance at publication.
Author Response
We thank the reviewer for their time and are glad they think the result is useful and relevant.
Based on the reviewer’s comments, we revised the results and explained as follows. We also use tracking revisions in MS so that reviewers can clearly understand the revisions we made based on the comments.
L175-185: What is the extent of your study? Did you perform any data reduction analysis to reduce the number of climatic variables you used here? Using all 19/23 for each scenario is not good at all! Beaumont et al. 2005 is one of the references I recall that evaluated this matter, but there are several others as I show you below. Later I saw you removed variables considering “highly correlated variables” (L185). What were the “highly correlated variables”? please explain.
RE: The extent of the study has been modified (Figure 4). We exclude highly correlated variables based on Variance Inflation Factor (VIF) values > 10 (section 2.3.1).
L 184-185 is modified as:
To reduce multicollinearity among the 24 variables (Table S1), the Variance Inflation Factor (VIF) implemented in the ‘usdm’ package in the R platform version 3.6.3 was used to exclude predictors with VIF values > 10 [42]
L188-198: There are better ways of building your training/testing subsets than random 70by30% partition.
RE: We calibrated models (training: 70/ testing :30%) according to Naimi and Araújo 2016.
L188-198 is modified as:
We employed an ensemble modelling approach to predict V. bicolor distribution using R with the ‘sdm’ package [42]. We selected five algorithms: generalized linear models (GLM), multiple adaptive regression splines (MARS), random forest (RF), maximum entropy (MaxEnt) and support vector machine (SVM). As all these models require background data (pseudo‐absence points), we generated a randomly sampling 5,000 locations across the study area. Models were evaluated using K-fold cross-validation with 10 folds and 10 replications for each algorithm. We projected each of the models using 70% of the training data and 30% for evaluation. The performance of Models was evaluated using threshold‐independent indices of area under the curve (AUC) of a receiver operating characteristic and the true skill statistic [43, 44]. The AUC ranged from 0.5–1, with a higher value denoting greater prediction accuracy. An AUC value ≥ 0.7 represents reliable prediction accuracy [45]. The relative importance of predictor variables was estimated using the function getVarImp of the ‘sdm’ package. Moreover, to obtain the consensus predictions, we used the function ’ensemble’ based on weighted TSS values. Finally, the output map represents the occurrence probability for each pixel. We extracted the pixels with maximum occurrence probability, representing probabilities more than 75% as suitable habitats [46]
L1194-198: AUC is a very poor metric to evaluate the model’s results, and it is highly criticized (Lobo et al. 2008; Jiménez-Valverde 2012, 2014). There are better metrics available in the market (Leroy et al. 2018).
RE: We have added a new indicator, the true skill statistic (TSS; section 2.3.1; 3.3).
L187-198: What was the threshold used to generate presence/absence maps? What were the future climatic scenarios used to generate future distributions and their sources? These are only two problems I could raise in a diagonal reading of the text. I am sure others may be found. Also, did you know that Maxent is very criticized if used alone, and several other improvements need to be taken care of Maxent’s settings and parametrization before you run it? Please see Muscarella et al. 2014 for some of these issues. I stress out that the authors need to acquire a good SDM text and check its M&M section to better structure the M&M section’s text.
RE: We extracted the pixels with maximum occurrence probability, representing probabilities more than 75% as suitable habitats (Giannini et al. 2012; section 2.3.1). We have not generated future distributions and have been revised in the text. We used ensemble modeling to reproduce the species distribution (section 2.3.1).
The RESULT is modified as:
The 24 initial variables were reduced to eight after applying the standard of VIF values > 10. Obtained values of TSS and AUC for five modelling algorithms indicated good predictive performance (TSS > 0.71 and AUC > 0.83), which can be considered an accurate validation. The results showed that the selected variables described the distribution of V. bicolor well. Among the eight environmental variables (Figure S1), the temperature seasonality (BIO4, 48.8% contribution) and temperature of driest quarter (BIO9, 12.8 % contribution) were two most important predictors determining the distribution (Figure S2).
The total amount of pixels with maximum occurrence probability (suitable habitats) covered 440 km2. This area was mainly distributed to cover 232 (52.0%) and 168 (38.0%) km2 in the counties of Miaoli and Hsinchu (Figure 4).

Round 2
Reviewer 1 Report
Dear authors. My comments are taken into account.
Author Response
We thank the reviewer for your appraisal of the work, and thank you for providing relevant literatures.
Reviewer 2 Report
The authors have done a fine job addressing all of my previous comments and those of other reviewers. I only have a few editorial suggestions below:
L32-33: Change “Thus, this study was to …” to “Thus, the aim of this study was to …”; or perhaps to “Thus, this study assessed the current status and traced origins …”
L54-55: Please revise this statement to eliminate grammatical errors or delete the sentence;
L62: idem;
L301-304: I’d suggest expanding this threat to agricultural and urban environments, beyond the native flora; I’d also suggest adding a few references to support this statement:
Pimentel, D., Zuniga, R. and Morrison, D., 2005. Update on the environmental and economic costs associated with alien-invasive species in the United States. Ecological economics, 52(3), pp.273-288.
Milosavljević, I., El-Shafie, H.A., Faleiro, J.R., Hoddle, C.D., Lewis, M. and Hoddle, M.S., 2019. Palmageddon: the wasting of ornamental palms by invasive palm weevils, Rhynchophorus spp. Journal of Pest Science, 92(1), pp.143-156.
Gaertner, M., Wilson, J.R.U., Cadotte, M.W. et al. 2017. Non-native species in urban environments: patterns, processes, impacts and challenges. Biol Invasions 19, 3461–3469.
Author Response
We thank the reviewer for your appraisal of the work again, and thank you for providing a few more relevant literatures. We will revise our article content following your suggestion. L32-33: Change “Thus, this study was to …” to “Thus, the aim of this study was to …”; or perhaps to “Thus, this study assessed the current status and traced origins …” Ans: Revised to "Thus, the aim of this study assessed the ......" L54-55: Please revise this statement to eliminate grammatical errors or delete the sentence; agree to delete the sentence Ans: Delete the sentence and the two references L62: idem; Ans: Thank you for the suggestion and we will revised the sentence to "Hornet has often become an example of an invasive species abroad. The yellow-legged horent Vespa velutina Lepeletier is a widely distributed species indigenous to Southeast Asia, and it invaded to Korea, Europe and Japan" L301-304: I’d suggest expanding this threat to agricultural and urban environments, beyond the native flora; I’d also suggest adding a few references to support this statement: Ans: Thank you for the suggestion and we will revised the sentence to "threat for the native flora [50], threat to the agricultural and urban environments [49–51] ,